# Parametric Damage Mechanics Empowering Structural Health Monitoring of 3D Woven Composites

**DOI:** 10.3390/s23041946

**Published:** 2023-02-09

**Authors:** Maurine Jacot, Victor Champaney, Francisco Chinesta, Julien Cortial

**Affiliations:** 1PIMM Lab, Arts et Metiers Institute of Technology, 155 Boulevard de l’Hôpital, 75013 Paris, France; 2Safran Tech, Department of Digital Sciences and Technologies, 1 Rue des Jeunes Bois, 78117 Châteaufort, France; 3CNRS@CREATE LTD, 1 Create Way, # 08-01 CREATE Tower, Singapore 138602, Singapore

**Keywords:** structural health monitoring, model order reduction, proper generalized decomposition

## Abstract

This paper presents a data-driven structural health monitoring (SHM) method by the use of so-called reduced-order models relying on an offline training/online use for unidirectional fiber and matrix failure detection in a 3D woven composite plate. During the offline phase (or learning) a dataset of possible damage localization, fiber and matrix failure ratios is generated through high-fidelity simulations (ABAQUS software). Then, a reduced model in a lower-dimensional approximation subspace based on the so-called sparse proper generalized decomposition (sPGD) is constructed. The parametrized approach of the sPGD method reduces the computational burden associated with a high-fidelity solver and allows a faster evaluation of all possible failure configurations. However, during the testing phase, it turns out that classical sPGD fails to capture the influence of the damage localization on the solution. To alleviate the just-referred difficulties, the present work proposes an adaptive sPGD. First, a change of variable is carried out to place all the damage areas on the same reference region, where an adapted interpolation can be done. During the online use, an optimization algorithm is employed with numerical experiments to evaluate the damage localization and damage ratio which allow us to define the health state of the structure.

## 1. Introduction

In aerospace engineering, structural health monitoring (SHM) represents a timely approach to evaluate in real time the integrity and safety of airplane components [1]. Based on this, SHM aims to gather insightful data from measured responses over time to identify in real time any unusual change in the structure behavior. Indeed, future aircraft will need to meet better standards for safety, dependability, supportability and lifespan. The application of multifunctional materials (such as composites) and technologies in essential aircraft structural components set strict criteria for the strength and structural integrity.

Advanced composite materials and more specifically three-dimensional (3D) woven composites have been widely used in different domains, including aerospace, civil engineering and other industries due to the benefits of a light weight, high ductility, corrosion resistance, and thermal resistance [2]. However, due to their complexity, they are susceptible to several types of structural damage, such as fiber breakage, matrix cracking and delamination [3]. At their onset, these damages are tiny and hardly noticeable to visual examinations. However, if left unrepaired, they might also result in catastrophes which, especially when airplanes are involved, result in a significant loss of human lives and financial losses. Therefore, it is important to monitor these structures.

More precisely, structural health monitoring (SHM) refers to an automated monitoring procedure that includes many new and advanced sensors and data processing techniques to assess the state of damage of a given structure of interest [4]. In the aerospace industry, monitoring systems are needed to evaluate the state of aircraft parts either during missions (diagnosis) or predicting their future behavior (prognosis). Different levels contribute to SHM: the first level is to detect if the structure is damaged and the type of damage; the second consists in locating the region where the structure is damaged and quantifying the size and severity of the damage.

SHM is a combination of a sensing system in the operational environment, a data acquisition system, data processing techniques and a damage detection procedure. The placement of sensors on the structure is fixed during design and manufacturing and defines how the monitoring will be accomplished. Data acquisition is performed by embedded technology connected to sensors to collect quantities of interest (e.g., strain, displacement). Data processing is combined with the damage detection algorithm. The aim is to interpret the data from the monitored structure to detect the onset of damage and assess its characteristics in real time to inform predicting maintenance procedures.

Depending upon the type of data processing, two approaches for damage detection have been developed: a “data-based” approach and a “model-based” approach. Both use an offline/online methodology. Actually, it offers an effective technique to carry out an SHM procedure.

Model-based approaches rely on the updating of a numerical model of the monitored structure. Inverse and optimization methods such as inverse finite element method (iFEM) [5], Kalman filter [6], optimization algorithms [7] or model correction and enrichment [8] are used to determine the structure’s health state through updated parameters. However, the inverse problem lead to costly calculations, which is not relevant for real-time SHM applications. Moreover, uncertainties and parameters variation may appear. Therefore, the data-based approach is becoming increasingly common.

The data-based approach is built entirely upon experimental data, but the main challenge is to construct a relevant offline database [9]. In most cases, it is difficult to obtain sufficient experimental data for a physical structure. Generally, experiments are limited and not available for a large number of test cases and possible damage conditions of the structure. To address this issue, in our approach, model-based and data-driven approaches are merged by incorporating a physics-based model commonly based upon the development of a finite element (FE) model of the considered structure to simulate a large number of damage configurations and to create an efficient dataset. They also enable access to properties and mechanical phenomena that are out of reach in experiments.

A powerful numerical technique to analyze the behavior of a 3D woven composite structure is the finite element method. It allows the prediction of the behavior of such structures and their collapse through the implementation of failure criteria. The FE method can be used to simulate the macroscale behavior of the structure under failure according to the different types of failure modes.

Several failure criteria models have been developed to investigate the effects of damage on the macroscale in 3D woven composites [10]. To study the aspects of stiffness reduction influencing material behavior, the Hashin criteria model is able to evaluate the influence of damaged modes on the material stiffness. They are associated with different damage modes using a distinct failure index [11] for fiber and matrix failure. The damage model must be able to simulate different types of damage at different locations so that the SHM system can perform damage detection and assessment.

However, one of the remaining challenges is the lack of predictive physical models to generate numerical solutions. These models are complex and must be solved using discretization techniques, such as, for example, the FE method. The construction of the offline dataset is therefore computationally expensive and makes their use incompatible with the requirements of a real-time response. Indeed, the offline database must be usable with a reduced computational time to be combined with a machine learning algorithm.

For this purpose, model order reduction (MOR) techniques for parametrized systems, such as the sparse proper generalized (sPGD) method [12], offer new opportunities to generate a consistent database that should be used in the offline training. The sPGD method provides a powerful tool to construct parametric solutions from a number of high-fidelity simulations according to a state-of-the-art design of experiments, while attempting to minimize the size of the sampling as much as possible. The sPGD method deals with the curse of dimensionality by the use of the concept of separate representation. Thus, an approximated function is expressed as a tensor product of separate terms resolved by a nonlinear solver. Using the approximated function, it is possible to evaluate any parameter value in the training intervals with a reasonable computational time. This method is already used for different applications, see [13,14].

However, when the chosen parameters are associated with the localization of the damage to the structure, the sPGD method is not accurate enough to provide valuable predictions. This is because the solutions are strongly influenced by the localization of the damage. To overcome this difficulty, we propose an adaptive sPGD using a change of variable in the parameters’ definition and an adapted interpolation procedure. Figure 1 presents the proposed SHM methodology relying on an offline/online strategy. In the offline phase, by the use of a consistent DoE and an adaptive sPGD method, a parametric solution of unidirectional fiber and matrix failure ratios and their localization on the structure is expressed. Then, in the online use, the parametric solution is combined with optimization methods to find the model parameters from measured data through a sensor network. By defining damage-related parameters in the dataset, such as fiber and matrix failure ratios and damage localization, the health state of the structure is determined.

The paper is organized as follows, Section 2 first details the construction of a damaged 3D woven composite model of an FOD panel using failure criteria and the FE method and then the construction of the reduced-order physics-based model using the classical sPGD method. In Section 3, the adaptive sPGD is presented using a change of variable and the adapted interpolation procedure. Last, Section 3.4 presents results of this approach on our case study. In Section 4, the inverse problem is defined and solved. Conclusion and remarks are finally discussed in Section 5.

## 2. Materials and Methods

Aerospace structures have employed 3D woven composites in a variety of applications, such as fan blades in the CFM International LEAP (Leading Edge Aviation Propulsion) engine (SAFRAN and GE) [15].

The fan is the first rotating element in contact with the air at the entrance of a turbojet engine. It consists of a number of blades arranged on a hub and rotating at the same speed as the rotor. Aircraft fan blades are manufactured using 3D fiber reinforced composites [16]. To demonstrate our methodology based on an offline/online strategy, we used numerical simulations of damaged blades to construct an offline database. The literature on numerical modeling of 3D woven composites reveals that several noteworthy investigations have been conducted in the past 20 years [17].

The foreign object damage panel (FOD panel) used in our case study is a representative substructure of the fan blade chord of dimensions 800 × 350 × 50 mm${}^{3}$ as shown in Figure 2 and it is used in a V&V (verification and validation) process by the engine OEM (original equipment manufacturer) to evaluate the designed and manufactured hybrid material regarding bird strike and its induced damage. The FOD panel is considered in the European project MORPHO (H2020 EU Project) [18].

### 2.1. Numerical Model

Three-dimensional woven composites exhibit a hierarchical structure. Three different scales can be distinguished by the representation of the structure of 3D woven composites. The microscale is the finest scale relevant to a continuum model; it describes the yarns as fiber filaments embedded in a matrix material. The mesoscale describes the woven architecture of the yarns, and the macroscale describes the material at the component level. The macroscale behavior of the material is influenced by the characteristics of the micro- and mesoscales.

In solid mechanics, the finite element method (FE) is usually employed to model and solve mechanical problems. The mechanics of the continuous model is represented by the subdivision into a number of elements whose behavior is represented by a finite number of parameters. For the sake of simplicity, the fan blade preform was assumed to be represented by an orthotropic material at the macroscale. The composite was assumed to be homogeneous and modeled as an elastic continuum.

The structural model of the fan blades used was based on the application of the linearized Lagrange equations. For the sake of simplicity and to focus on methodologies, we restricted our analysis to the static case.

Consider a domain $\Omega \subset R^{3}$; the global equilibrium equation reads: (1)KU=F,
where *F* and *U* are, respectively, the vectors of nodal forces and displacements in three dimensions.

The solution of the global problem presented in Equation (Equation 1) is obtained by assembling the elements in the FE model, following the rules of the usual FE method.

The global force vector *F* results from the assembly of the so-called elementary force vectors. The elasticity matrix **K** is specific to the properties of the considered material and results from the assembly of all element stiffness matrices.

In intrinsic notation, the elasticity tensor establishes a multilinear application between the stress and strain tensors. For anisotropic materials, Hooke’s law can be written as:(2)σ=Hϵ,
where σ and ϵ are the macrostress and macrostrain tensors, respectively, expressed in a vector form and H is the macroscale orthotropic elastic matrix that reads:(3)$$H=\left[\begin{array}{cccccc} \frac{1-\nu_{yz}\nu_{zy}}{E_{y}E_{z}\Delta } & \frac{\nu_{yx}+\nu_{zx}\nu_{yz}}{E_{y}E_{z}\Delta } & \frac{\nu_{zx}+\nu_{yx}\nu_{zy}}{E_{y}E_{z}\Delta } & 0 & 0 & 0\\ \frac{\nu_{xy}+\nu_{xz}\nu_{zy}}{E_{z}E_{x}\Delta } & \frac{1-\nu_{zx}\nu_{xz}}{E_{z}E_{x}\Delta } & \frac{\nu_{zy}+\nu_{zx}\nu_{xy}}{E_{z}E_{x}\Delta } & 0 & 0 & 0\\ \frac{\nu_{xz}+\nu_{xy}\nu_{yz}}{E_{x}E_{y}\Delta } & \frac{\nu_{yz}+\nu_{xz}\nu_{yx}}{E_{x}E_{y}\Delta } & \frac{1-\nu_{xy}\nu_{yx}}{E_{x}E_{y}\Delta } & 0 & 0 & 0\\ 0 & 0 & 0 & G_{xy} & 0 & 0\\ 0 & 0 & 0 & 0 & G_{xz} & 0\\ 0 & 0 & 0 & 0 & 0 & G_{yz} \end{array}\right],$$
(4)$$\Delta =\frac{1-\nu_{xy}\nu_{yx}-\nu_{yz}\nu_{zy}-\nu_{zx}\nu_{xz}-2\nu_{xy}\nu_{yz}\nu_{zx}}{E_{x}E_{y}E_{z}}.$$

The engineering constants, the moduli $E_{x}$, $E_{y}$ and $E_{z}$, the Poisson ratios $\nu_{xy}$, $\nu_{xy}$, $\nu_{xz}$ and $\nu_{yz}$ and the shear moduli $G_{xy}$, $G_{xz}$ and $G_{yz}$ express the linear elasticity in an orthotropic material.

### 2.2. Damage Mechanics of Composite Structures

In the present study, a constitutive model based on continuum damage theory was employed to perform quasi-static damage simulations of the 3D woven composite fan blade. This choice was made for modeling the impact of the damage to the structure by locally decreasing the stiffness.

Fiber and matrix failures are the most frequent 3D woven composite failures, most of which take place internally and are hardly perceptible. They should be detected early to prevent catastrophic structural failures since they can significantly reduce the performance of composite materials.

By considering that the damage is already initiated, the existence of microcracks reduces the ability of the damaged structure to support loads. Consider a region, where an effective section $A_{0}$ is reduced to $A_{D}$ because of the presence of numerous microcracks. The undamaged region is then subjected to higher stresses. This gives rise to the idea of effective stress, which is the stress that applies in the undamaged region. The nominal stress σ and the effective stress $\sigma^{\prime }$ are related by a damage variable *D* that quantifies the area reduction.
(5)$$\sigma^{\prime }=\frac{1}{1-D}\sigma ,\;\;\;\;\;\;\mathrm{with}\;D=\frac{A_{D}}{A_{0}}.$$
According to the 3D formulation, in a matrix form, the effective stress in terms of nominal stress reads:(6)$$\sigma^{\prime }=D\sigma ,\;\;\;\;\;\;\mathrm{with}\;D=W^{-1},$$
where W is the damage factor matrix given by:(7)$$W=\left[\begin{array}{cccccc} \left(1-D_{11}\right) & 0 & 0 & 0 & 0 & 0\\ 0 & \left(1-D_{22}\right) & 0 & 0 & 0 & 0\\ 0 & 0 & \left(1-D_{33}\right) & 0 & 0 & 0\\ 0 & 0 & 0 & \left(1-D_{44}\right) & 0 & 0\\ 0 & 0 & 0 & 0 & \left(1-D_{55}\right) & 0\\ 0 & 0 & 0 & 0 & 0 & \left(1-D_{66}\right) \end{array}\right].$$

For 3D woven composites, the damage variables are based on Hashin’s theory. The damage variables $D_{11}$, $D_{22}$ and $D_{33}$ represent the damage modes of the fiber, matrix and shear in the fiber direction of a fiber strand while $D_{44}$ and $D_{66}$ represent the combination of fiber breakage and transverse matrix cracking (out-of-plane); finally, $D_{55}$ represents transverse matrix fractures (in-plane and out-of-plane). When the material is damaged, the stress in Equation (Equation 2) is updated based on the damage stiffness matrix and the strain $\epsilon^{d}$. The constitutive equation reads:(8)$$\sigma^{d}=H^{d}\epsilon^{d},$$
where $\sigma^{d}$ and $\epsilon^{d}$ are the damaged stress and strain tensors, respectively, expressed in a vector form. $H^{d}$ is the damaged elasticity matrix as follows:(9)$$H^{d}=HW.$$

### 2.3. Model Order Reduction Using Sparse Proper Generalized Decomposition

Model order reduction (MOR) has gained popularity in recent years, because of its advantage in significantly reducing the computational cost while minimizing a loss of accuracy. The objective is to define a simplified representation of the evolution of physical systems. In the context of the FE method, it corresponds to the use of a very reduced number of degrees of freedom.

#### 2.3.1. Parametric High-Fidelity Solution

A parametric solution depends on $N_{D}$ features (parameters) within a parametric space D that could represent geometric parameters, material parameters or any other parameter involved in a generic model.
(10)$$f\left(p^{1},\ldots p^{N_{D}}\right):D\subset R^{N_{D}}\to R.$$

Computing parametric solutions based on surrogates (metamodels or response surfaces) requires defining a design of experiments (DoE) to work with. Generally, the number of points for the sampling grows exponentially with the number of dimensions $N_{D}$. A method commonly used to achieve a reasonably accurate random distribution covering D while reducing the number of samples is the Latin hypercube sampling (LHS). Then, if we consider the parameters $p^{1},p^{2},\ldots ,p^{N_{D}}$ involved in the vector p∈D, the sampling that constitutes the DoE results in the parameters choice $p_{b}$, where b=1,…B, and *B* is the number of computed solutions trying to cover the parametric space as much as possible. Then, for each set of parameters, a high-fidelity simulation is computed.

In the present case study, to demonstrate our methodology, four parameters $N_{D}=4$ related to damage were considered. The aim of the SHM framework consists of the detection, localization and characterization of damage. For that purpose, we considered the damage model presented in Section 2.2 combined with the FE solver to emulate a damage on the blade. That damage led to a stiffness reduction at a certain location of the structure. Two parameters were chosen to demonstrate the proposed methodology and characterize the damage severity, $D_{11}$ and $D_{22}$, respectively. In our structure, the FOD panel was subjected to a longitudinal tensile force. The most likely initial failure in this configuration is a fiber failure and it can be followed by other failure mechanics such as a matrix failure. Finally, in order to assess the localization of the damage on the structure, two other parameters characterized the damage location, $\bar{x}$ and $\bar{y}$. The size of the damage ($\bar{L}\times \bar{l}$) was constant in our study. That is, once the FE model was created, the damage variables became parametric.

A single *instance* of the data was comprised of a set of elements solution of the FE model e=1,…E
*E* being the total quantity of collected data at elements, represented by the strain in direction x, $\epsilon_{xx}$ such as $f\left(D_{11_{b}},D_{22_{b}},{\bar{x}}_{b},{\bar{y}}_{b}\right)=\left[\epsilon_{{xx}_{b}}^{1},\epsilon_{{xx}_{b}}^{2},\ldots ,\epsilon_{{xx}_{b}}^{E}\right]\in R^{B\times E}$ The model was trained with B=250 samples according to the Latin hybercube sampling.

#### 2.3.2. Sparse Proper Generalized Decomposition

The sparse proper generalized decomposition (sPGD) aims at approximating such a function by a sum of products of one-dimensional functions. This has an advantage, since it allows one to reduce a high-dimensional problem into a series of several one-dimensional problems computed consecutively until convergence. In our work, the function was constructed using data collected at the chosen elements of the FE model that were represented by their two coordinates *x* and *y* as the computation proceeded. The dataset was split into two parts, the first part to train the model and a second part to test it.

Considering an objective function *f* in the parametric space D that we try to approximate, the separate representation of the approximated solution is written as:(11)$$f\left(p^{1},\ldots ,p^{N_{D}}\right)\approx \tilde{f}\left(p^{1},\ldots ,p^{N_{D}}\right)=\sum_{m=1}^{M}U_{m}\prod_{d=1}^{N_{D}}\psi_{m}^{d}\left(p^{d}\right)\in R^{E},$$
where $\tilde{f}$ is the approximation of *f*, $U_{m}$ is a column vector which has *E* rows, *M* denotes the number of PGD modes and $\psi_{m}^{d}$ is the one-dimensional function for modes *m* and dimension *d*.

The precise form of the functional $\psi_{m}^{d}\left(p^{d}\right)$ was obtained by first projecting it in a standard approximation basis and by employing a greedy algorithm. That is, once the approximation to order M−1 was completed, we searched mode *M* as follows:(12)$$f\left(p^{1},\ldots ,p^{N_{D}}\right)\approx \tilde{f}\left(p^{1},\ldots ,p^{N_{D}}\right)=\sum_{m=1}^{M-1}U_{m}\prod_{d=1}^{N_{D}}\psi_{m}^{d}\left(p^{d}\right)+U_{M}\prod_{d=1}^{N_{D}}\psi_{M}^{d}\left(p^{d}\right),$$
Functions $\psi_{m}^{d}\left(p^{d}\right)$ with m=1,…,M were expressed from a standard approximation basis $N_{m}^{D}$ and coefficients $a_{m}^{d}$:(13)$$\psi_{m}^{d}\left(p^{d}\right)=\sum_{i=1}^{Nf}N_{i,m}^{d}a_{i,m}^{d}={\left(N_{m}^{d}\right)}^{T}a_{m}^{d}$$
where $N_{f}$ represents the number of degrees of freedom of the one-dimensional functions and $N_{m}$ is the vector containing the shape functions.

To construct the one-dimensional functions $\psi_{m}^{d}\left(p^{d}\right)$, several options can be employed, such as polynomial basis functions, piecewise linear shape function, splines, kriging, etc. To demonstrate the methodology, Legendre polynomial basis functions were chosen.

Finally, the approximation error minimization was:(14)$$\tilde{f}\left(p^{1},\ldots ,p^{N_{D}}\right)=\arg\min_{f^{*}}||f-f^{*}{\left|\right|}_{2}^{2}=\arg\min_{f^{*}}\sum_{b=1}^{B}||f\left(p_{b})-f^{*}(p_{b}\right){\left|\right|}_{2}^{2}.$$

#### 2.3.3. Numerical Simulations

The FE analysis was performed at the macroscale by using Abaqus software, to determine macrostrains in the first direction $\epsilon_{xx}$ due to applied load and damage modes. The structure was subjected to a tensile force F=1000
*N* in the *x* direction on one of its sides while the displacement vanished on the opposite side. The mesh was comprised of 129 thousand elements of C3D8R (8-node hexahedra with reduced integration) elements that consisted of 175 thousand nodes. Figure 3 shows the finite element model considered in the numerical simulations.

The considered material properties of the 3D woven composite preform are given in Table 1. The intervals related to each parameter are given in Table 2 with $\bar{L}$ = 40 mm and $\bar{l}$ = 20 mm.

The FE model was applied using the high-fidelity solver in Abaqus software for B=250 damaged configurations according to the DoE. To demonstrate the ability of the method to reconstruct different types of damage on the structure, we chose to show six simulations in the test database. For each of them, the damage was observed in the area with the maximum strain. Figure 4 shows the $\epsilon_{xx}$-component of the strain on the top-view *xy*-plane for the damage configurations reported in Table 3.

#### 2.3.4. Reduced Basis Approximation

We applied the sPGD methodology presented in Section 2.3.2 using a classical polynomial basis. M=100 modes were used, and the training dataset was comprised of the first 230 high-fidelity solutions and the test dataset of the last 20 high-fidelity solutions. To train the model, we used 1728 elements of the 2D model presented in Figure 4. Figure 5 compares the reference solution *f* and the values predicted using the sPGD regression using a polynomial basis $\tilde{f}$. The blue and red points represent the training and test dataset, respectively, in the parametric domain.

The perfect prediction is indicated by the diagonal line. We can notice that the regression failed to predict the right values of the strain $\epsilon_{xx}$. Very certainly the regression underperformed due to the solution localization, which was difficult to approximate by considering a usual polynomial approximation.

## 3. Adaptive Sparse Proper Generalized Decomposition

### 3.1. Motivation

We observed that the sPGD model was not able to approximate the parametric solution when damage was located in different areas along the surface of the structure. To ensure accurate results, the proposed approach applied to the initial dataset (DoE) a change of variable. With this technique, it was possible to center the coordinate system around the damage location, making the interpolation easier to obtain. For all the damaged configurations of the DoE, the parameters related to the damage localization $\bar{x}$ and $\bar{y}$ were subtracted from their coordinates as $\bar{x}-x$, $\bar{y}-y$. The result of the change of variable for the damage configurations proposed in Table 3 and Figure 4 is shown in Figure 6. All the damages were centered in the middle of the FOD panel surface.

The sPGD method represents a powerful tool to approximate parametric solutions, but functions $\psi_{m}^{d}\left(p^{d}\right)$ cannot be approximated using the classical polynomial basis when the involved parameters induce a solution localization, as is the case for the parameters defining the location of the damaged area. Thus, a valuable route to enhance accuracy consists in approximating the associated functions related to the damage localization using a piecewise cubic interpolation. Polynomials are too smooth ($C^{\infty }$) and tend to grow very quickly when approaching the border of the domain (except for the constant polynomial). The parameter functions in this case need to capture a localized perturbation in the center of the domain and be close to a constant near the borders. Using piecewise cubic functions is a good compromise, because it allows the representation of localized behaviors.

### 3.2. Modified Dataset Construction

A change of variables was applied to center functions involved in the damage location: $\bar{x}$ was replaced by $\bar{x}-x$ and $\bar{y}$ was replaced by $\bar{y}-y$.

Consider a new function that we are trying to approximate that depends on different dimensions in the new parameter space $D^{*}\left(D_{11_{b}},D_{22_{b}},{\left(\bar{x}-x\right)}_{b},{\left(\bar{y}-y\right)}_{b},S\right)\subset R^{4}\times N$, where S is the element id of the considered mesh in Section 2.3.4. The sampling presented in Section 2.3 was used. The vector output was expressed as $f(D_{11_{b}},D_{22_{b}},{\left(\bar{x}-x\right)}_{b},{\left(\bar{y}-y\right)}_{b},S)$ = $\left[\epsilon_{{xx}_{1}}^{1},\ldots ,\epsilon_{{xx}_{1}}^{E},\epsilon_{{xx}_{2}}^{1},\ldots ,\epsilon_{{xx}_{2}}^{E},\ldots ,\epsilon_{{xx}_{B}}^{E}\right]\in R^{(B\times E)\times 1}$ . The first four functions were approximated using piecewise functions, while the functions involving element ids were functions of a discrete variable. Figure 7 presents the adaptive sPGD methodology. In the first part, high-fidelity simulations were computed using the DoE, then centered and approximated as a function using a greedy algorithm and a piecewise cubic approximation basis.

### 3.3. Regression Using Piecewise Cubic Functions

We sought to approximate $\psi_{m}^{d}\left(p^{d}\right)$ in the new parameter space $D^{*}$, defined by $N_{f}$ piecewise cubic functions. For that, we considered a piecewise cubic function $N_{i,m}^{d}\left(p^{d}\right)\in C^{2}$ for each $i=0,1\ldots ,N_{f}-1$.
(15)$$\psi_{m}^{d}\left(p^{d}\right)=\sum_{i=1}^{N_{f}}N_{i,m}^{d}a_{i,m}^{d}={\left(N_{m}^{d}\right)}^{T}a_{m}^{d}.$$

In our work, the number of functions $N_{f}$ was a hyperparameter of the problem and could be adjusted to obtain the best compromise between underfitting and overfitting, through means such as a cross-validation. Figure 8 shows an example of the piecewise cubic basis functions for $N_{f}=10$ functions in the interval [0,1000].

Then, using these regressions, the sPGD method presented in Section 2.3.2 was implemented as previously explained. Equation (Equation 12) was computed using a greedy approximation.

### 3.4. Results

The result obtained by the adaptive sPGD is presented in Figure 9. The blue and red points represent the training and test datasets, respectively, in the parametric domain. We observed that the improved method provided more accurate predictions of $\epsilon_{xx}$ than the classical sPGD results shown in Figure 5. The adaptive sPGD method allowed us to approximate the parametric solution which contained localized solutions. The precision of the learning stage was essential because it was then used in the online stage. Figure 10 shows a comparison between real values presented in Figure 4 and predicted values using the adaptive sPGD of the $\epsilon_{xx}$-component of the strain on the top view xy.

The relative errors between the classical sPGD method and the adaptive sPGD in the test dataset are compared in Table 4. Each method’s error in the test dataset was determined as follows:(16)$$E_{rr}=\frac{∥{\vec{f}}_{B_{test}}-{\vec{\tilde{f}}}_{B_{test}}{∥}^{2}}{∥{\vec{f}}_{B_{test}}{∥}^{2}}.$$
where ${\vec{f}}_{B_{test}}$ is the reference solution evaluated on the test dataset $B_{test}=20$ and ${\vec{\tilde{f}}}_{B_{test}}$ is the approximate solution.

We observed that by choosing an adequate approximation basis that fitted each one-dimensional function combined with a change of variable, the adaptive sPGD method had a lower approximation error compared to the traditional sPGD method with a classical polynomial basis.

## 4. Online Parameter Estimation

In structural health monitoring settings, the problem of structural parameter estimation from measurements collected by only a few sensors is a well-known challenge. By using the offline parametric database proposed in Section 2, the objective was to solve an inverse problem. An inverse problem describes how knowledge about a parameterized physical system can be obtained from experimental data, that is, the relationships between model parameters and data. For that purpose and for the sake of simplicity, we assumed that the experimental data were computed numerically and collected in a set of points; the sensor locations are shown in Figure 11 in orange and correspond to some elements $e_{k}$, with *k* the sensor set. All measured data were stored in a scalar form in the vector $f^{M}$. Thus, the optimization problem was:(17)$${\left(p^{1},\ldots ,p^{N_{D}}\right)}^{M}=\arg\min\sum_{s=1}^{k}{\left|{\tilde{f}}_{s}\left(p^{1},\ldots ,p^{N_{D}}\right)-f_{s}^{M}\right|}^{2},$$
where ${\left(p^{1},\ldots ,p^{N_{D}}\right)}^{M}$ are the estimated parameters and ${\tilde{f}}_{s}$ is the approximated function chosen at the sensors’ location.

To solve the minimization problem (Equation 17), the Nelder–Mead simplex method was employed [19]. The Nelder–Mead method is a common numerical method to find the minimum/maximum of a multidimensional objective function. The algorithm used was provided by the Python optimization library SciPy [20]. Parameters related to the damage mode were initialized randomly from a uniform distribution while the parameters related to the damage localization were initialized by the coordinates of the element $e_{k_{max}}$, with the highest strain value $\epsilon_{xx}$ in the measurements.

The estimated parameters from the minimization problem corresponding to the damage configurations in Figure 4 are presented in Table 5. By the use of the parametric approximation and the optimization algorithm, we observed that the estimated parameters were broadly in line with the input parameters, which showed the robustness of the adaptive sPGD.

Our approach made it possible to localize the damage on a structure by a good estimation of the parameters $\bar{x}$ and $\bar{y}$. Both parameters had a strong influence on the solution since we had to adapt the classical approach to successfully capture these dependencies.

Moreover, we highlighted that the damage parameter $D_{11}$ had a stronger influence on solutions than $D_{22}$. Indeed, solutions were reconstructed correctly with the estimated parameters, even if $D_{22}$ was not accurate enough. Several reasons can be discussed, for example, to only observe $\epsilon_{xx}$. It would be then necessary to also learn $\epsilon_{yy}$ and to place sensors in the *y* directions. However, the model would be harder to learn and the structure would be further complexified.

## 5. Conclusions

This work proposed a structural health monitoring method relaying an offline/online strategy. To overcome the lack of experimental data, model-based approach and data-driven approaches were combined to create an offline database of a damaged FOD panel by the use of a physics-based numerical model. The dataset consisted of the most probable 3D woven composite failures when the FOD panel was subject to a longitudinal tension: a fiber failure and a matrix failure. Moreover, different locations of the damage were taken into account.

To alleviate the computational cost of the numerical model and the interpolation difficulties related to localized solutions, an adaptive sparse proper generalized was proposed that allowed us to evaluate the influence of the parameters faster without the need to perform simulations. Thus, once the parametric space was created, damages became parametric, making it possible to evaluate them in real time. The reference solution was approximate using the concept of separate representation, which dealt with the curse of dimensionality. Then, during an online phase, numerical sensor’s measurements from the FOD panel to be diagnosed were generated and combined with the parametric approximation and an optimization algorithm to define the health state of the structure from a parameter estimation.

The proposed method allowed us to evaluate the influence of many parameters in the strain response of the structure with a reduced computational time and provided a relevant damage detection of the structure. This approach could be exploited using experimental data as soon as the numerical model is developed. Despite the proposal to improve the traditional sPGD method using a change of variable and an adequate approximation basis, another interpolation procedure could be exploited to handle the difficulties of localized solutions.

## Figures and Tables

**Figure 1 sensors-23-01946-f001:**
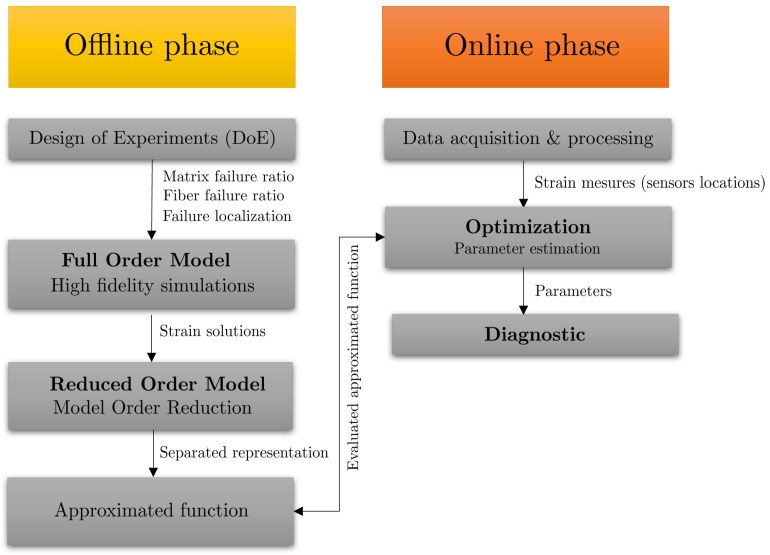
Offline/online-based SHM strategy.

**Figure 2 sensors-23-01946-f002:**
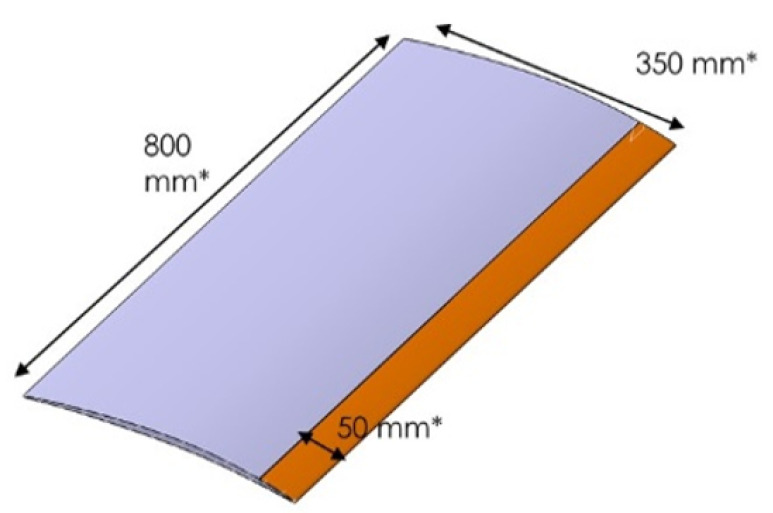
Schematic diagram of the geometry of the fan blade preform considered in the H2020 EU project. The 3D woven composite FOD panel is the structure shown in grey. The titanium leading edge (not considered in this study) is the orange part.

**Figure 3 sensors-23-01946-f003:**
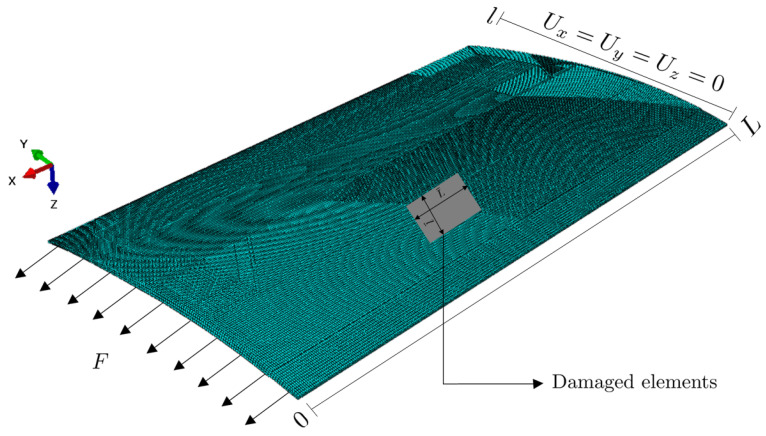
Finite element (FE) model of the structure and boundaries conditions.

**Figure 4 sensors-23-01946-f004:**
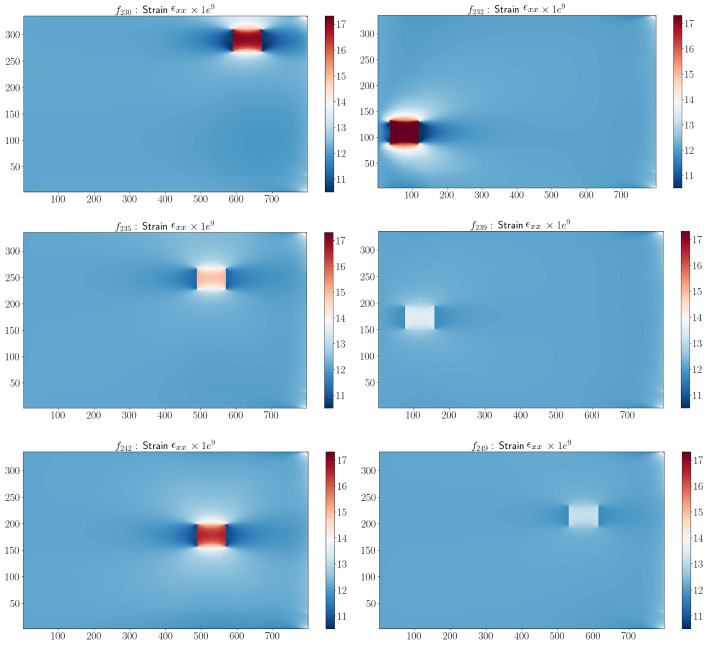
$\epsilon_{xx}$-component corresponding to damage configurations in Table 3.

**Figure 5 sensors-23-01946-f005:**
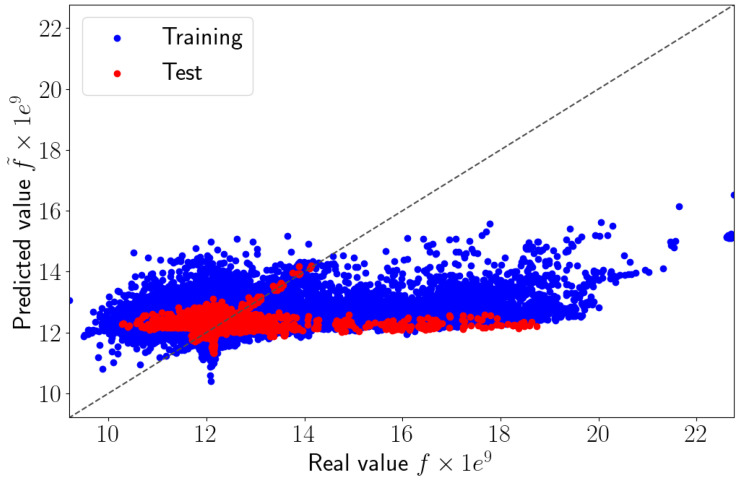
Predicted values versus real values using sPGD method.

**Figure 6 sensors-23-01946-f006:**
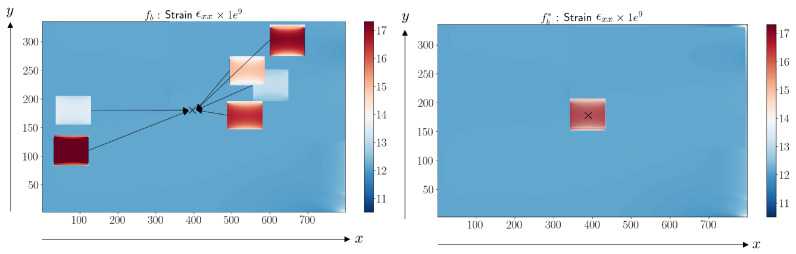
Centering damages prior to applying interpolation.

**Figure 7 sensors-23-01946-f007:**
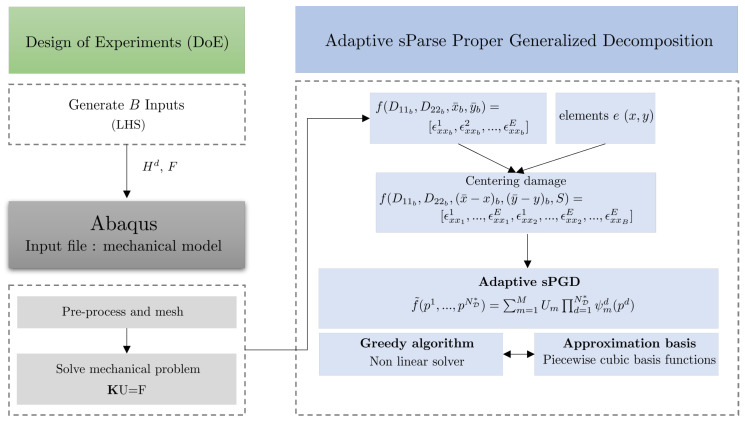
Adaptive sPGD methodology.

**Figure 8 sensors-23-01946-f008:**
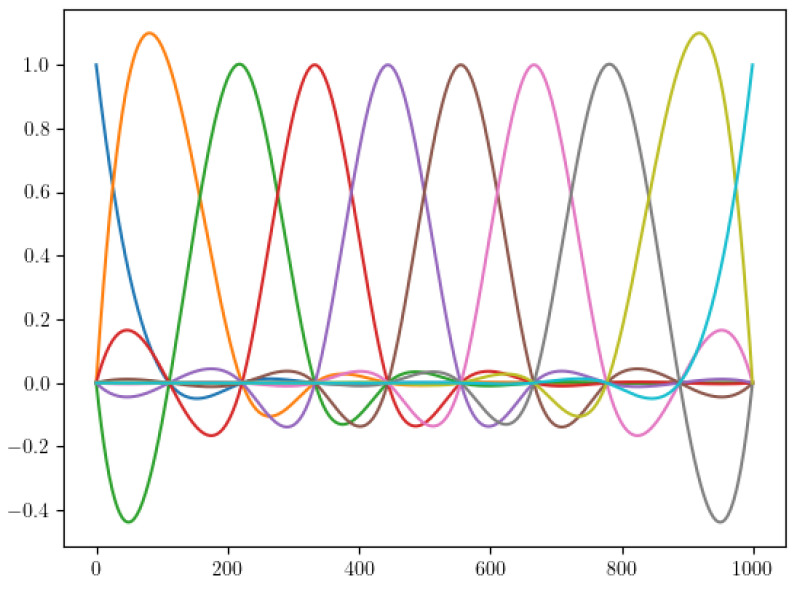
Piecewise cubic basis functions with $N_{f}$ = 10.

**Figure 9 sensors-23-01946-f009:**
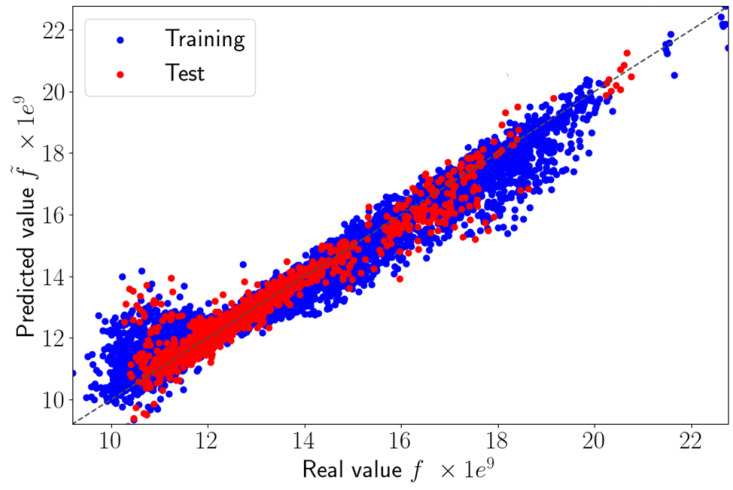
Predicted values versus real values using adaptive sPGD.

**Figure 10 sensors-23-01946-f010:**
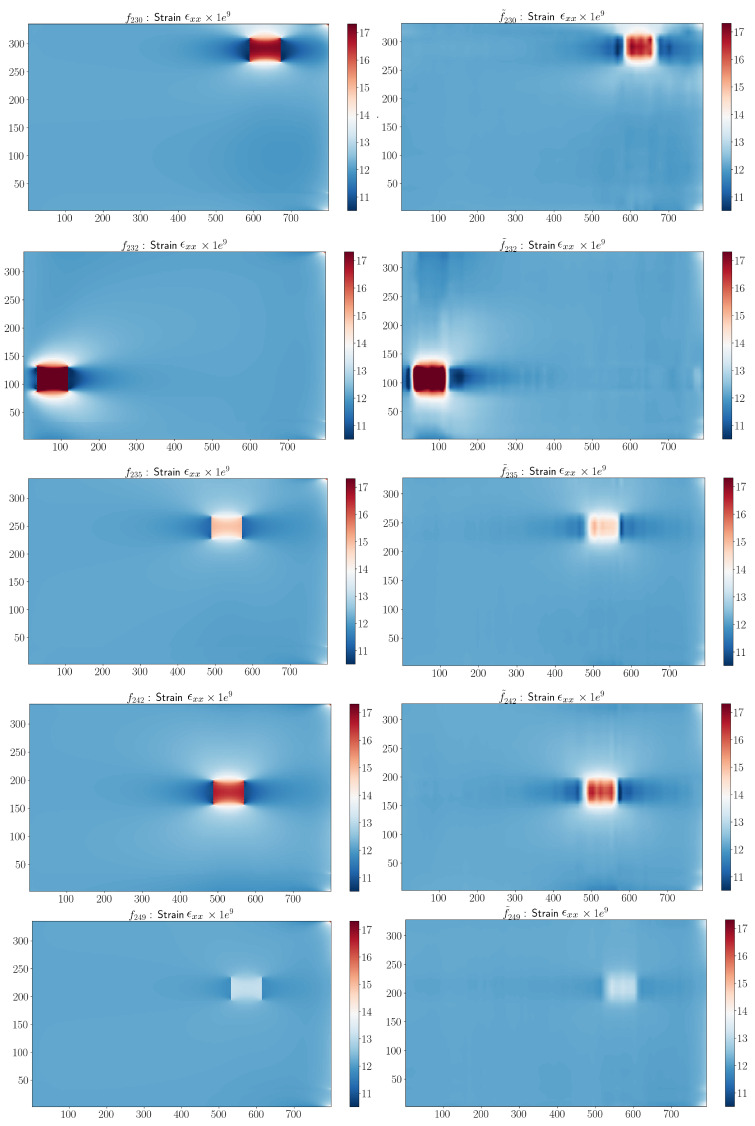
Comparison between real and predicted $e_{xx}$-component corresponding to the damage configurations in Table 3.

**Figure 11 sensors-23-01946-f011:**
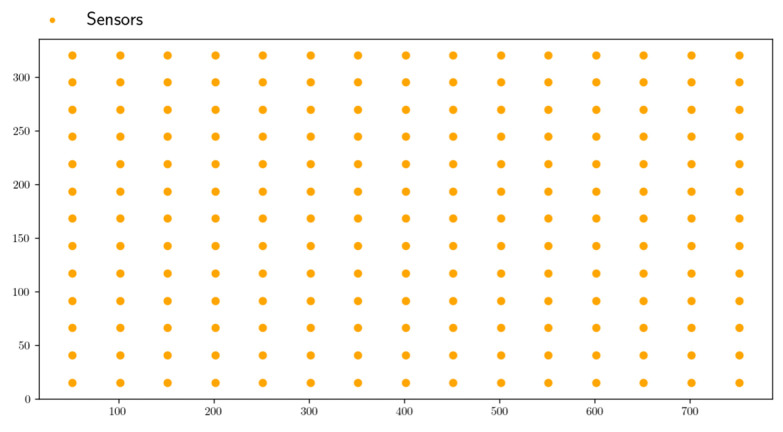
Sensors’ location.

**Figure 12 sensors-23-01946-f012:**
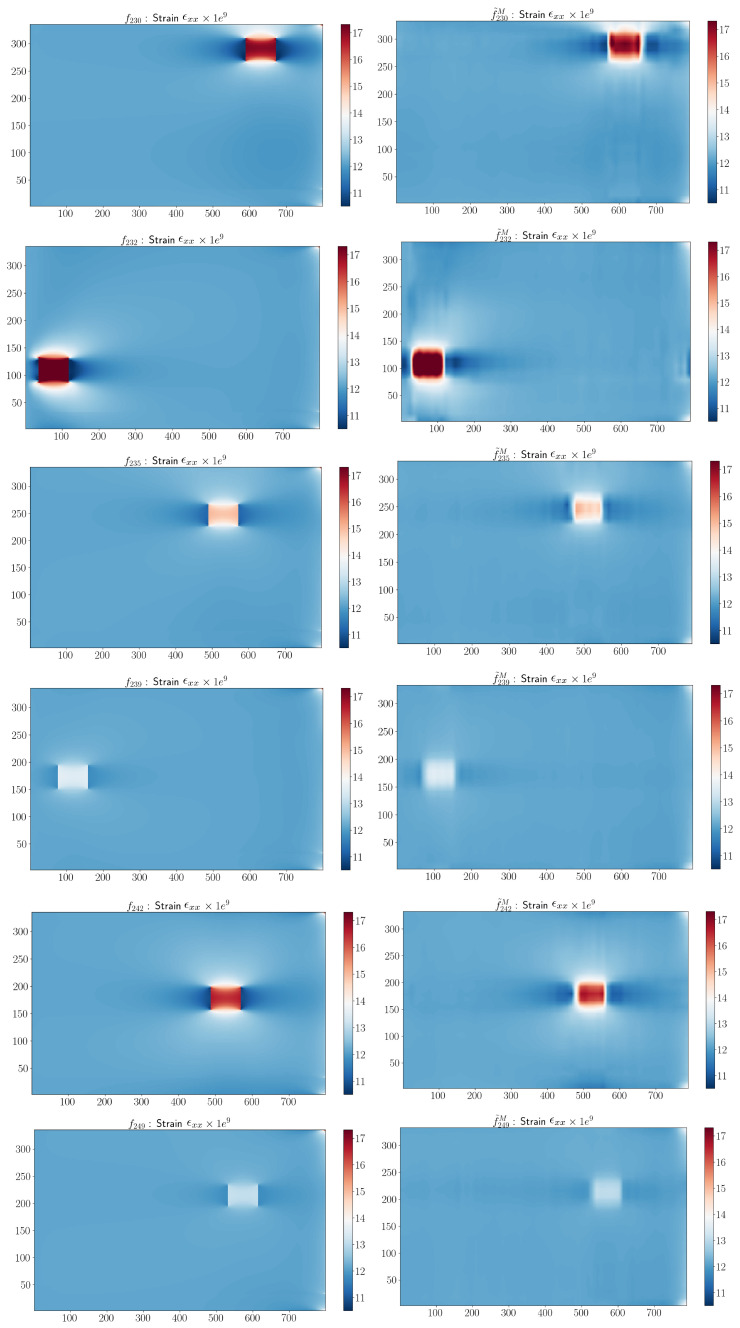
Comparison between real and estimated $\epsilon_{xx}$-components corresponding to the damage configurations in Table 5.

**Table 1 sensors-23-01946-t001:** Material properties.

Property and Symbol	Value
Elastic modulus $E_{x}$	82.46 (GPa)
Elastic modulus $E_{y}$	37.77 (GPa)
Elastic modulus $E_{z}$	5.31 (GPa)
Poisson’s ratio $\nu_{xy}$	0.17
Poisson’s ratio $\nu_{yz}$	0.32
Poisson’s ratio $\nu_{xz}$,	0.31
Shear modulus $G_{xy}$	6.03 (GPa)
Shear modulus $G_{yz}$	2.24 (GPa)
Shear modulus $G_{xz}$	2.27 (GPa)

**Table 2 sensors-23-01946-t002:** Parametric space.

Parameters	Minimum	Maximum
Fiber damage amplitude $D_{11}$	0	0.5
Matrix damage amplitude $D_{22}$	0	0.5
Damage position $\bar{x}$	0	$L-\bar{L}$
Damage position $\bar{y}$	0	$l-\bar{l}$

**Table 3 sensors-23-01946-t003:** Input parameters corresponding to strain $\epsilon_{xx}$ shown in Figure 4.

Parameters	$f_{230}$	$f_{232}$	$f_{235}$	$f_{239}$	$f_{242}$	$f_{249}$
$D_{11}$ (-)	0.3783	0.4922	0.2523	0.1293	0.3467	0.1011
$D_{22}$ (-)	0.4576	0.4942	0.1269	0.0439	0.1383	0.3097
$\bar{x}$ (mm)	671.97	117.03	570.88	64,20	568.80	604.9
$\bar{y}$ (mm)	270.95	89.92	228.11	153.84	158.84	195.10

**Table 4 sensors-23-01946-t004:** Relative errors obtained by the sPGD and adaptive sPGD methods.

Methods	Relative Errors
sPGD–polynomial basis functions	0.0288
Adaptive sPGD–piecewise cubic functions	0.005

**Table 5 sensors-23-01946-t005:** Estimated parameters corresponding to the strain $E_{xx}$ shown in Figure 12.

Parameters	$f_{230}^{M}$	$f_{232}^{M}$	$f_{235}^{M}$	$f_{239}^{M}$	$f_{242}^{M}$	$f_{249}^{M}$
$D_{11}$ (-)	0.3872	0.4840	0.2410	0.1305	0.3464	0.1085
$D_{22}$ (-)	0.3450	0.4331	0.1100	0.0325	0.1586	0.2005
$\bar{x}$ (mm)	678.73	115.15	560.88	64.380	563.46	610.47
$\bar{y}$ (mm)	271.70	85.02	228.34	154.56	159.08	196.03

## Data Availability

Data available under request.

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
