# Peer review of "Parametric Damage Mechanics Empowering Structural Health Monitoring of 3D Woven Composites"

_sensors, 2023, doi:10.3390/s23041946_

Round 1
Reviewer 1 Report
Numerical models optimized by an adaptive sParse Proper Generalized decomposition were developed for real-time structural health monitoring (SHM) of composite fan blade in this manuscript; and the structural parameters of the blade was inversed by minimizing the error between the measurements and the simulation results of the model with the Nelder-Mead simplex method. However, there are many points need to be revised before considering for publication in this journal. Its research topic and results were not clearly introduced in the manuscript. For example, the reason and execution process of the nonlinear regression for real time structural health monitoring were not explained clearly; the introduction should be revised to help potential read understand the meaning of the research; whether the developed method can be verified by experiments to show it has engineering application potential? Followings are some other problems:
1. The application performance of the developed approach should be highlighted in the abstract; and what structural parameters were used in the SHM.
2. The full expression of PGD should be used in the keywords.
3. The different between the model-based approaches and the data based approached was not clearly in the lines 52-71. In this manuscript, the researched was developed based on the optimized numerical models. Why it was classified as a data-based approach.
4. In line 53, iFEM? Lines 63, 93, [9,10], [14,15]; the FE method in the line of 106.
5. In line 123, 800[mm]×350[mm]×50[mm]->’ 800×350×50mm3’;
6. Figures, 1, 3~10, should be descripted in the relevant paragraphs;
7. If the model simulation was not based on Eq. (1), it can be removed from the manuscript.
8. σ and ε were redefined in Eqs. (3) and (9).
9. As expressed in Eqs. (6)~(10), many parameters were used in the quasi-static damage simulation of the 3D woven composite fan blade. Why the four parameters, D11, D22, x and y, were chosen in the research?
10. In lines 219~225, the expressions of the standard approximation basis and the constructed one-dimensional functions are necessary in the work.
11. Can you provide a figure to clearly show out the establishment of numerical models optimized by an adaptive sParse Proper Generalized decomposition?
12. In line 229. 1000 N;
13. Table 4 can be descripted in relevant paragraph.
14. The implementation process of the Nelder-Mead simplex method?
15. Finally, the advantage and shortage of the developed approach should be discussed.
Author Response
We thank the reviewers for their pertinent remarks, that we addressed in the revised version to improve its quality. The introduction has been revised to explain the meaning of our research and two tables have been added to clearly explain the developed methodology. Corrections are written in red in the revised version. At this moment, experiments from the European Morpho project are not available to verify the developed method. However, the method will be validated when they are available with the FOD panel. In this article, we therefore used test simulations as experimental data. Indeed, the longitudinal strain can be easily measured using a sensor network. Also, we removed the Figure 1(b) which does not provide additional information on our structure.
Please see in the attachment, the response to Reviewer 1 comments point by point.
Kind Regards,
Maurine Jacot, Victor Champaney, Francisco Chinesta, Julien Cortial
Reviewer 2 Report
The sensitivity analysis of the numerical model may be included highlighting significant and minor parameters
Were any experimental studies conducted to validate the model?
Author Response
We thank the reviewers for their pertinent remarks, that we addressed in the revised version to improve its quality. The introduction has been revised to explain the meaning of our research and two tables have been added to clearly explain the developed methodology. Corrections are written in red in the revised version. At this moment, experiments from the European Morpho project are not available to verify the developed method. However, the method will be validated when they are available with the FOD panel. In this article, we therefore used test simulations as experimental data. Indeed, the longitudinal strain can be easily measured using a sensor network.
Please see the attachment, the response to Reviewer 2 comments, point by point.
Kind regards,
Maurine Jacot, Victor Champaney, Francisco Chinesta, Julien Cortial

Round 2
Reviewer 1 Report
Althrough almost all comments were revised reasonably, I am still confused about the title, because the tittle can not show out the topic and innovation of the work.
Therefore, it is suggested to use a more appropriate title to highlight the reseach content, which will help to attract the attention of researchers.
Author Response
We thank the reviewers for their pertinent remarks, that we addressed in the revised version to improve its quality. The tittle has been revised.
Kind regards,
Maurine Jacot, Victor Champaney, Francisco Chinesta, Julien Cortial